## Research Article

coastal Indigenous peoples; food security; constraints; limits; climate change adaptation

**Corresponding author:**
Eranga K. Galappaththi;
Email: eranga@vt.edu

# Solvable constraints and unsolvable limits to global climate adaptation in coastal Indigenous food security

Eranga K. Galappaththi [ID], Sithuni M. Jayasekara [ID], Chrishma D. Perera [ID], Gayanthi A. Ilangarathna [ID] and Hannah Garbutt [ID]

Department of Geography, Virginia Polytechnic Institute and State University, Blacksburg, VA, USA

## Abstract

Coastal systems are a major source of food for Indigenous communities. Climate change poses a high risk to coastal communities' food security. Successful climate change adaptation practices are essential to ensure food security among Indigenous peoples. Yet, limits and constraints challenge climate change adaptation practices. Our study seeks to identify these limits and constraints in the context of food security among coastal Indigenous peoples. We performed a global scale systematic literature review using 155 scholarly articles to examine the constraints and limits to climate adaptation in the coastal food security and Indigenous peoples' context. The three research questions are as follows: (i) What are the key constraints? (ii) What are the limits? (iii) What are the ways of overcoming the constraints? First, we found that, globally, the main constraints to adapting to climate change in coastal food security settings are related to governance, institutions and policies. Second, most limits are soft, to be solved, compared to hard limits on coastal systems. Third, we unveiled ways of overcoming the constraints, such as restoring coastal food system resilience, improving food accessibility and building the adaptive capacity of Indigenous peoples. The findings of the study provide valuable insights for policy-makers, researchers and other relevant stakeholders involved in decision-making regarding coastal food security in the climate change adaptation context.

## Impact statement

Our research highlights the vulnerabilities and strengths of coastal Indigenous communities concerning climate change, especially food security. By pinpointing and examining the barriers to climate adaptation practices, the study offers practical insights that hold relevance both locally and globally. The main findings emphasize that governance issues, inefficiencies within institutions and gaps in policy are the key limitations, while the majority of adaptation challenges are considered "soft," suggesting that there are possible solutions through innovation and collaboration. Proposals such as restoring the resilience of coastal food systems, improving food access and strengthening adaptive capacities are outlined as feasible strategies to deal with these challenges. Focusing on coastal Indigenous communities – who are particularly vulnerable to climate change – the study underscores their specific reliance on aquatic food systems and the urgent threats they encounter. This research enhances the understanding of how historical colonial impacts and current governance issues contribute to food insecurity in coastal communities. Policymakers, researchers and stakeholders engaged in climate change adaptation can gain significantly from the insights provided by the study. By presenting an approach for managing and addressing constraints while exploring the boundaries of "soft" limits, the research equips decision-makers with effective tools to tackle food security issues in fragile coastal areas. Beyond its immediate focus, the findings have wider implications for global sustainable development. They promote collaborative efforts across sectors to enhance social-ecological systems, creating a future where Indigenous knowledge systems and traditional practices are integral to strategies for climate resilience.

## Introduction

Coastal communities are highly sensitive to climate impacts. Climate change events such as the rising sea level, rising water tables and increasing saltwater intrusion incursions affect coastal communities in a variety of ways, such as loss of land, destruction of infrastructure and reduction in income (Dolan and Walker, 2006; Abas et al., 2017). For instance, globally, coastal Indigenous peoples are 15 times more dependent on aquatic food than non-Indigenous peoples (Cisneros-Montemayor et al., 2016). According to Cisneros-Montemayor et al. (2016), coastal Indigenous peoples consume an average of 2.1 million metric tons of seafood, which is equal to around 2% of

the global yearly commercial fish catch. Thus, in this context, climate change has a significant impact on coastal communities' food security. Food security is a situation where all people always have access to enough good, safe food to lead healthy, active lives (Alonso et al., 2018). Commonly observed root causes of food security issues include changing the food web, which has unpredictable effects on fish stocks, and increasing the risk of invasions and the spread of vector-borne diseases that threaten coastal communities' food security (Cochrane et al., 2009).

Coastal Indigenous peoples experience food insecurity issues at an alarming rate. For example, northern Canadian Indigenous peoples experience food insecurity at a rate two to six times higher than that of average Canadian households (De Position, 2016). Nearly 50% of households belonging to First Nations communities residing on reserves experienced high levels of food insecurity, according to the findings of Batal et al. (2021). The transition from a high-protein, low-carbohydrate diet to a high-sugar and high-fat diet of processed foods is often associated with food insecurity (Kuhnlein et al., 2004, 2013). Indigenous peoples' dietary patterns have changed over time due to changes in their lifestyles, such as spending less time on the land and abandoning traditional techniques in fishing, resulting in food insecurity issues (Usher, 2002; Islam and Berkes, 2016). Colonialism disrupted coastal Indigenous food systems via environmental damage, land loss, ecological impacts of disasters, restricted access to healthy environments, compromised nutrition and increased exposure to pollutants (Evans-Campbell, 2008; Walters et al., 2011; McKinley, 2023a,b). Additionally, climate change is one of the most significant factors influencing the food security of coastal Indigenous peoples in terms of food availability, accessibility, utilization and stability (Shafiee et al., 2022).

Climate adaptation is essential in responding to risks associated with coastal communities. Yet, adaptation has its own limits and constraints (Carter, 2011). According to Morrison and Pickering (2013), consideration of limits to adaptation to climate change will be important in decision-making about adaptation strategies. Understanding the limits to climate change helps determine the feasibility of climate change adaptation strategies, ascertain the temporal effectiveness of adaptation strategies based on climate change predictions, enhance the understanding of societal values and facilitate prioritization of adaptation strategies (Morrison and Pickering, 2013). Hence, successful adaptation requires a proper understanding of the limits and constraints of adaptation to climate change, which is a greater concern to researchers (Moser and Ekstrom, 2010; Thomas et al., 2021; Bertana et al., 2022).

The term "limit" is defined as "the point at which an actor's objectives or system's needs cannot be secured from intolerable risks through adaptive action" (Klein et al., 2015, 907). Limits are mainly categorized as soft or hard. According to Thomas et al. (2021), a soft limit is one in which adaptation options are currently unavailable but could be available in the future, while a hard limit is an option in which additional adaptations can no longer be made. Adger et al. (2009) assigned limits to climate change adaptations into four categories as follows: (i) biophysical limits, (ii) economic limits, (iii) technological limits, and (iv) social limits. "Barriers or constraints are referred to as obstacles that can be overcome with concerted effort, creative management, change of thinking, prioritization and related shifts in resources, land uses and institutions" (Moser and Ekstrom, 2010, 22027). Thomas et al.'s (2021) study delineated eight types of constraints: economic, social/cultural, human capacity, governance/institutions and policy, financial, information/awareness/technology, physical and biological (Table 1). Adaptation constraints and adaptation limits differ from one another; while constraints can be eliminated, the limit is a

**Table 1.** Definitions of the types of constraints

| Type | Description |
| --- | --- |
| Economic | Existing livelihoods, economic structures and economic mobility |
| Social/cultural | Social norms, identity, place attachment, beliefs, worldviews, values, awareness, education, social justice and social support |
| Human capacity | Individual, organizational and societal capabilities to set and achieve adaptation objectives over time, including training, education and skill development |
| Governance/institutions and policy | Existing laws, regulations, procedural requirements, governance scope, effectiveness, institutional arrangements, adaptive capacity and absorption capacity |
| Financial | Lack of financial resources |
| Information/awareness/technology | Lack of awareness of, or access to, information and technology |
| Physical | Presence of physical barriers |
| Biological | Temperature, precipitation, salinity, acidity and intensity and frequency of extreme events, including storms, droughts and winds |

*Source:* Thomas et al. (2021, 3).

threshold at which drastic modifications are required with no alternative options available (Moser and Ekstrom, 2010; Barnett et al., 2013, 2015; Dow et al., 2013). To allow for timely and efficient adaptation to climate change, understanding and managing the limits and constraints is essential (Biesbroek et al., 2013; Thomas et al., 2021).

The expanding body of literature provides a foundation for analyzing and quantitatively synthesizing how constraints and limits are currently being faced and framed at a global scale (Thomas et al., 2021). Sietsma et al. (2021) found that adaptation research has increased by 20.6% per year from 2009 to 2019. While extensive research has highlighted the general impact of climate change on global food security, less attention has been paid to coastal Indigenous communities' specific adaptive capacities and unique vulnerabilities to food insecurity (Gregory et al., 2005; El Bilali, 2020; Berrang-Ford et al., 2021). Additionally, there remains a lack of knowledge about constraints and limits to climate adaptation focusing especially on coastal food security among Indigenous peoples (Galappaththi et al., 2024). Our study will address this knowledge gap. The study conducts a systematic literature review to advance understanding of the documented constraints/barriers and limits associated with coastal climate change adaptation in the "Indigenous food security context." The three research questions are (i) What are the key constraints to adapting to climate change? (ii) What are the limits to adapting to climate change? (iii) What are the most commonly documented ways of overcoming the constraints?

Our study makes a distinctive contribution to the existing scholarship by examining the documented and experiential limits to adaptation within coastal Indigenous communities. It particularly focuses on how these limits hinder the communities' ability to preserve traditional food systems in the face of climate change. Addressing these gaps is vital not only for enhancing the resilience of Indigenous communities but also for enriching the global understanding of sustainable adaptation practices that can be applied across various social-ecological contexts.

## Methods

We used a systematic literature review approach to examine the constraints and limits to climate adaptation in the coastal food security and Indigenous peoples context. The systematic literature review approach employs a stepwise process to search, filter, review, analyze, interpret and summarize findings from numerous publications on a specific area of interest (Pati and Lorusso, 2018). This approach has been applied to multiple subjects, such as environmental policy, climate adaptation and health (Gopalakrishnan and Ganeshkumar, 2013; Macura et al., 2019; Shaffril et al., 2020). Figure 1 explains the steps used in the systematic literature review in a flow diagram.

To conduct our search, we first identified the following four databases: (i) Web of Science (WoS), (ii) Scopus, (iii) Cab Direct and (iv) AGRICOLA by ProQuest. WoS and Scopus are large, multidisciplinary databases offering access to a comprehensive and vast array of published studies related to climate change and food security. CAB Direct is dedicated to agriculture and associated sciences, whereas AGRICOLA centers on agriculture and associated areas. This makes them especially appropriate for research involving coastal Indigenous communities involved in agricultural activities in the context of environmental science. To ensure the feasibility and manageability of data extraction, we have not included additional databases. We developed search strings to find publications linking food security, climate change adaptation, coastal communities and Indigenous peoples to systematically identify relevant publications that focus on the intersection of these interconnected themes. Our search strings were database-specific. However, we included search terms ("coast*") AND ("communit*," OR "village*," OR "rural*") AND ("climat*") AND ("chang*") AND ("adapt*") AND ("knowledge*") AND ("Indigenous OR local OR traditional") AND ("food*") OR ("Subsistence OR fish*" OR "hunt*") commonly in all databases with database-specific adjustments. The database-specific search strings that we developed and the number of publications obtained are given in Supplementary Table S1. We searched for this string in the title, abstract and keywords. Looking through the title, abstract and keywords helped us maintain focus and relevance by concentrating on brief recaps of the main subject of the paper (title), detailed summaries of goals and outcomes (abstract) and essential topics clearly specified by the authors (keywords). We conducted our search in March 2023 and did not limit it to any particular discipline, time duration or article type. Given the target audience and language translation limitations, we looked for articles published in English.

In our next step, we consolidated the articles obtained from each database into one Excel sheet. To identify and remove duplicates, we used the digital object identifier. After the duplicates were removed, our initial data set consisted of 170 articles. The 170 articles were extracted into a new Excel sheet for an initial screening. Our research team consisted of five members. The lead researcher has expertise in this area, and the four other researchers had previous experience with systematic literature review. Four researchers, excluding the lead researcher, conducted an initial screening of the articles by screening about 44 articles individually. All five researchers met weekly to discuss issues and progress. Our inclusion and exclusion criteria were that the article should focus on human adaptation for food security in changing climates. Using the guiding criteria in Supplementary Table S2, we excluded any article that did not fit the context of food security, humans and climate change. The number of excluded articles per each criterion is listed in Supplementary Table S2. Following the initial screening, the four screeners undertook a comprehensive quality check. Here, each of the screeners examined the others' quality checking. Specifically, each screener went through another's screening process, selected 25% random articles from the total articles of 170 and verified whether the screeners had performed their duties correctly. Discrepancies that emerged during this quality-checking phase were resolved through collaborative discussions. To ensure rigor and consistency, the lead researcher carried out the ultimate round of quality checking.

Upon conclusion of the screening process, a total of 155 articles had been selected for coding. This signified the exclusion of 24 articles from the original pool of 170. Our coding process encompassed the systematic collection of data concerning constraints and limits to climate change adaptations along with ways of overcoming the constraints in the context of coastal Indigenous peoples (Supplementary Table S3). We performed manual coding with the participation of a team of four members. Then, we checked the quality of the coding. For this, we distributed the coding articles among ourselves and verified their quality. Each member randomly selected 10% of the articles that had been designated to others, reviewed those articles and determined whether they had been coded correctly. Utilizing the screened data, we constructed the descriptive results and presented them via various modes of representation, such as percentages, numerical counts, graphs and maps, to vividly portray our findings. For clarity in presenting the descriptive findings, we rounded the calculated percentages to the nearest whole number.

Within the framework of this study, we engaged in both manifest and latent content analyses (Krippendorff, 2018). These analytical techniques allowed us to identify underlying themes and, thus, enabled the exploration of connections between the diverse variables and apparent patterns within the data. We accomplished the first objective by taking percentages of each constraint across regions. Similarly, to meet the second objective, we calculated the

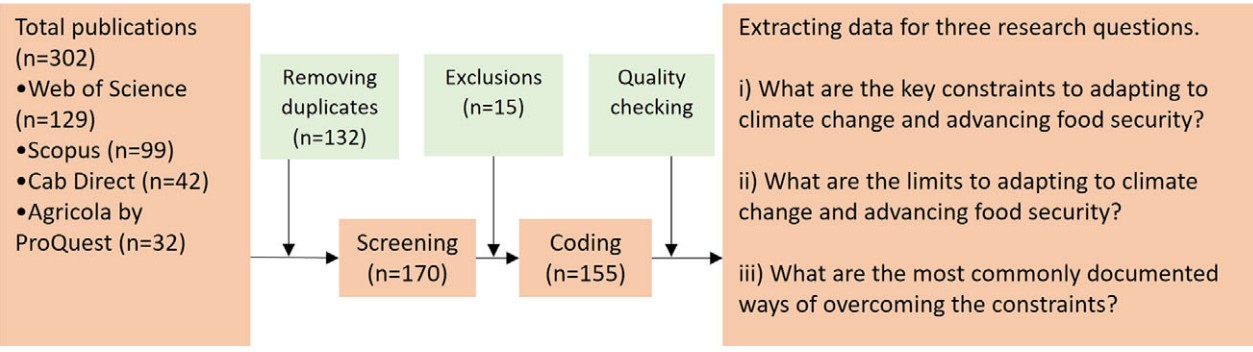

**Figure 1.** Flow diagram of the steps followed in the systematic review.

percentages of soft limits and hard limits across regions. For the third research objective, we identified themes to recognize the constraints and ways of overcoming those constraints.

## Results

The research conducted a global-level systematic literature review within six continents, that is, North America (33%, *n* = 54), South America (3%, *n* = 5), Asia (26%, *n* = 42), Africa (12%, *n* = 19), Europe (10%, *n* = 16) and Oceania (17%, *n* = 27), while covering a time span ranging from 2009 to 2023. The publications as reported by the journals included marine policy (5%, *n* = 8), ecology and society (4%, *n* = 6), ocean and coastal management (4%, *n* = 7), climate risk management (3%, *n* = 4), climate change management (3%, *n* = 5) and climate change (3%, *n* = 5). The first authors of the study were predominantly affiliated with countries such as Canada (21%, *n* = 33), Australia (20%, *n* = 31), the United States (17%, *n* = 27), India (6%, *n* = 9) and South Africa (4%, *n* = 6). Three percent of the authors (*n* = 5) were primarily affiliated with institutions such as McGill University, University of Victoria (3%, *n* = 5), Rhodes University (3%, *n* = 4), Simon Fraser University (3%, *n* = 4) and University of the Sunshine Coast (3%, *n* = 4).

### *Types of key constraints*

Adaptation constraints are the factors that make it harder to plan and implement adaptation actions; they are also referred to as obstacles or barriers (Mechler et al., 2020). Figure 2 illustrates the nine types of categories of constraints: economic, social/cultural, human capacity, governance, financial, information/awareness,

physical, biological and other across the continents. The study specifically focuses on how these constraints influence the food security of coastal Indigenous peoples. We found that governance/institutions and policies are the primary constraint (15%, *n* = 106) to adapting to climate change in coastal food security settings. Galappaththi et al. (2021) highlighted that power imbalances among fishers can affect the resilience of small-scale fisheries systems. The imbalance in power creates unequal access to fishing resources, which, in turn, leads to overexploitation and ultimately reduces food availability for the community. In Zanzibar (an island that is part of the United Republic of Tanzania), formal institutions lack the capacity to administer efficient, long-term monitoring systems of environmental change, which will exacerbate vulnerability and delay climate change adaptation and, in turn, disrupt the food supply (Zhang and Bakar, 2017). Whitney and Ban (2019) also referred to the lack of government actions and policies as a constraint to climate change adaptations in coastal British Colombia, indicating an increasing need to research the background of constraints associated with governance, institutions and policies that promote efficient adaptation.

Moreover, there has been a more frequent occurrence of barriers to adaptation due to societal, cultural and economic factors (14% each, *n* = 96). Van Putten et al. (2014) found that fishing communities with strong cultural inertia will not try to change their fishing practices with the changing environmental conditions, reflecting a social/cultural constraint. Biological constraints indicate a lower frequency for each of the eight categories. For example, the development of harmful algae blooms has led to increased food insecurity because of reduced food access for coastal communities (Gianelli et al., 2021).

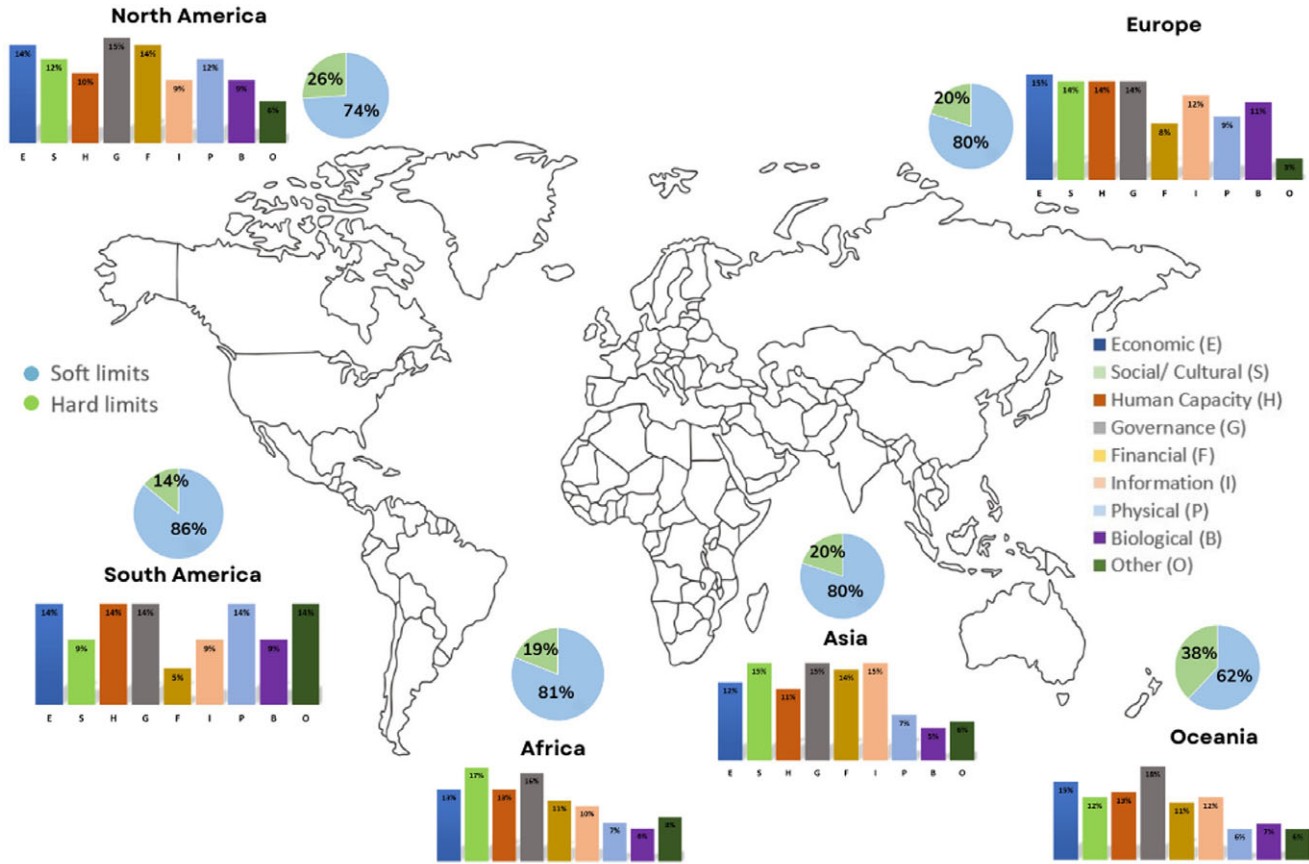

**Figure 2.** Types of constraints and limits across continents.

The study identified some other barriers. Among these, educational, communication and health barriers play a vital role. Inabilities to read and write and limitations on the communities' language literacy can be categorized under both educational and communicational constraints (Fischer et al., 2022; Putiamini et al., 2022). Health-based barriers, such as the spread of disease, have also been found to be a constraint in coastal areas (Costello et al., 2009; Cochrane et al., 2019). Examples were found of infrastructure barriers, such as small areas of cultivated land and loose housing structures (Hasan and Kumar, 2022). Gender-based barriers, such as differences in the connection between food security and gender, have been identified by Savage et al. (2020) and Das and Mishra (2022).

In North America and Oceania, constraints related to governance account for a significantly higher proportion, that is, 15% ($n = 35$) and 18% ($n = 18$), respectively, while in South America, economic, human capacity, governance, physical and other are shown to have a higher percentage (14% each, $n = 3$). In contrast, a higher proportion in the African continent (17%, $n = 15$) is accounted for by social and cultural constraints. Meanwhile, 15% ($n = 31$) of the Asian continent is characterized by social/cultural, governance and informational constraints. Considering the limits across continents, soft limits prevail over hard limits in all six continents. Table 2 shows the evidence of constraints and adaptation responses to food security and who adapts in coastal communities.

### Limits to coastal adaptation and food security

The findings of the study show that most limits are soft limits with a 78% chance of being solvable, as opposed to hard limits in coastal systems. For example, Dagar and Tewari (2017) highlighted that if land degradation continues for the next 25 years, global food production will be limited due to increasing demand coupled with an increasing coastal population. The problems that land degradation creates – for example, declining soil fertility and soil productivity, and increasing salinity (especially in coastal regions) – will lead to yield losses. As a result, food availability will decrease with rising demand from an increasing population. Shaffril et al. (2017) suggested that fishers possess a strong attachment to their occupation that prevents them from adopting alternative income-generating activities. This strong attachment leads to negative consequences, especially when bad weather conditions in the future limit marine resources and the number of days available to be at sea. Poverty will increase and the purchasing power of fishers and

**Table 2.** Evidence of constraints to coastal adaptations regarding food security

| Type of constraint | Example of constraint | Implication of the constraint on food security | Adaptation strategies used (and by whom) | References |
|---|---|---|---|---|
| Economic | Population growth, rising incomes and changing consumption patterns will limit the availability of food, energy and water by at least 50%, 45% and 30%, respectively, by 2030. | Contributing to heightened vulnerability to food insecurity by restricting food accessibility. | Protecting and restoring soil fertility and rehabilitating degraded lands (coastal communities, community-based organizations). | (Dagar and Tewari, 2017) |
| Social/cultural | Addictions to alcohol, cigarettes, gambling and drugs negatively influence people's material possessions and motivations. | Leading to health consequences that limit fishers' ability to work, thereby exacerbating food insecurities. | Prohibiting alcohol sales within the community; Paulatuk is a damp community (coastal community leaders, community-based organizations). | (Lede et al., 2021) |
| Human capacity | The livelihoods of fishers are restricted in seasons when fish catches are poor. | Seasonal variations can undermine fishers' ability to build financial resilience, thus limiting livelihood opportunities and enhancing vulnerability to food insecurities. | In seasons when fish catches are poor, fishers shift their livelihoods; men look for other opportunities and women might cook and sell food (fishers). | (Sowman and Raemaekers, 2018) |
| Governance/ institutions and policy | The Tamil Nadu government's caste system (called "Other Backward Castes") restricts women's freedom of movement. | Restricting women's ability to work to preserve their livelihoods and reducing resilience to climate change adaptation, which leads to food insecurity. | Revitalizing policies and traditional caste systems (government authorities). | (Axelrod et al., 2022) |
| Financial | A lack of financial capabilities exists with regard to the implementation of climate initiatives. | Reinforcing the risk, as fishers cannot invest in essential infrastructure, such as fishing gear and equipment, thus leading to reduced productivity among fisheries. | Acquiring funding from relevant stakeholders, such as government, non-government, local, and international organizations (fishers). | (Kettle et al., 2018) |
| Physical | Repeated flooding will damage ponds while decreasing resilience regarding climate change adaptations. | Reducing the fishing yield due to the destruction of habitats. | Reconstructing flood-damaged ponds (fishers). | (Putiamini et al., 2022) |
| Information/ awareness and technology | Information on hydro-meteorological hazards is lacking when young people move away from rural areas to find income. | Leading to reduced catches of fish because of their increasing vulnerability to climate change. | Documenting and properly transmitting traditional knowledge and practices across generations (elders in the coastal communities). | (Irvine et al., 2020) |
| Biological | Increased incidences of harmful algal blooms (HABs) limit the consumption of shark liver and sardine heads. | Increasing vulnerability to food insecurity, as HABs have a detrimental effect on fish biology, leading to reduced quality. | Proper functioning of water quality checking (local authorities). | (Cochrane et al., 2019) |

families will be reduced to such an extent that they will face a restricted ability to obtain food.

Irreducible uncertainties reduce the resilience of small-scale fisheries systems as an unsolvable hard limit in the global north and south (Galappaththi et al., 2021). Rural small-scale fisheries are facing uncertainties because they depend on economic and market systems to maintain local fishing activities. Fisheries are subject to higher market price fluctuations because of the resulting uncertainties (e.g., unpredictability in weather patterns), which affects the accessibility to food for people who lack purchasing ability. The issue of uncertainties in scientific understanding and among practitioners (coastal managers and planners) has also been studied as a limit for climate change adaptations in coastal British Columbia (Whitney and Ban, 2019). In Asia and Oceania, due to the challenges and uncertainties associated with monitoring and evaluating adaptation, many ecosystem-based adaptation projects have not assessed their approach or defined their success, which has led to greater levels of uncertainty surrounding predicted future climatic changes (Giffin et al., 2020). Such a situation will exacerbate coastal communities' vulnerability to climate change, leading to food insecurity through the loss of livelihoods and income, reduced fish catches and increased market prices of fish. Table 3 represents the evidence of limits and adaptation responses to food security and who adapts.

## Overcoming constraints to coastal food security

Our study recognized ways to overcome constraints in the context of coastal climate change adaptation. Communities in the Circumpolar North are facing food security issues because access to, and the availability of, wildlife species are declining (Ford et al., 2021). Food security issues are also accelerating due to changes in the migration timing of fish such as Arctic char (*Salvelinus alpinus*) resulting from climate change impacts. This reflects the fact that physical constraints have negative effects on the food security of coastal communities in the Arctic. Supplementing this argument, as a physical constraint, increasing ocean temperature influences fish movement and harmful algal blooms (HABs; Cochrane et al., 2019). Regarding this scenario, the authors recommend focusing on developing marine heatwave indicators, establishing temperature thresholds and establishing a HAB index. While HABs have severely affected fishers in the Southwest Atlantic Ocean, these fishers remain optimistic about their future (Gianelli et al., 2021). Cochrane et al. (2019) studied how food security could be ensured by the creation of new supply chain opportunities for fishing communities negatively affected by climate change. Our study found that Indigenous peoples (NiVanuatu) experience persistent poverty in their communities. To overcome this constraint, we suggest that subsistence farming be promoted by demonstrating garden plots and establishing community-based reservation areas

**Table 3.** Evidence of limits to coastal climate change adaptations in the food security context

| Type of limit | Evidence | Implication of the constraint on food security | Adaptation strategies used (and by whom) | References |
|---|---|---|---|---|
| Soft | Temperature variability limits coastal crop production (i.e., scorching of leaves, influence on the timing of flowering, desiccating the crop and damaging pollen). | Increasing vulnerability as temperature variability reduces crop yields, leading to food insecurity. | Cultivating varieties resistant to climate change (coastal farmers, researchers, crop breeders, community-based organizations). | (Egerer et al., 2019) |
| Soft | Shellfish harvesting is limited because of harmful algal blooms. | Decreasing shellfish yield leads to increased food insecurity. | Monitoring and maintaining water quality (government and non-government authorities, research institutions). | (Gianelli et al., 2021) |
| Soft | Fish prices are limited because fishers are the price takers who are increasingly bound to processors/buyers who set the fish prices. | Fluctuating market prices negatively influence the food accessibility of people with low purchasing power. | Facilitating bargaining power, which can improve trade fisheries (supply chain actors, fishers, fishing associations, community-based organizations). | (Metcalf et al., 2015) |
| Soft | If land degradation continues until 2050, global food production will be reduced. | Lagging agricultural productivity leads to reduced food security as the population grows. | Increasing food production through sustainable land management and conservation agriculture by 2050 (farmers in coastal communities, agricultural extension officers, government agencies). | (Dagar and Tewari, 2017) |
| Soft | Planning of adaptation needs in fisheries will fail when municipal plans have a narrow focus. | Insufficient planning leads to inappropriate adaptation practices, which negatively affects fish production and productivity. | Focusing on adaptation plans on certain climate impacts (e.g., sea-level rise), considering as a combination rather than isolated (municipal administrations, research and academic institutions) | (Maltby et al., 2023) |
| Soft | Nontechnically supervised seawall construction has accelerated beach erosion, limiting fish-producing and processing practices (i.e., seaweed drying and fish landing). | Depleting fish stocks stems from habitat degradation of breeding grounds. | Constructing proper seawalls (government agencies, coastal management authorities). | (Zhang and Bakar, 2017) |
| Hard | Fishing associations have a limited ability to enforce regulations, as fishers share access rights based on territorial user rights for fishers. | Allocating resources and food equally is difficult, resulting in increasing vulnerability to food insecurity. | Because this is a hard limit, there are no adaptation responses. | (Andrachuk and Armitage, 2015) |

**Table 4.** Ways of overcoming constraints to food security in coastal systems

| Context | Constraint | Solution | Reference |
|---|---|---|---|
| Food production | Productivity and survivorship of food crops are severely influenced by extended periods of high heat or drought. | Optimizing irrigation practices by increasing the frequency of watering while taking into consideration the temperature variabilities. | (Egerer et al., 2019) |
| Food accessibility | Unexpected weather changes reduce access to food and culturally significant species. | Facilitating access to emerging technologies (i.e., media or networks) to increase an individual's ability to obtain healthy food sources amid environmental fluctuations of coastal food systems. | (Lemelin et al., 2010) |
| | Access to marine food species is declining because of the co-occurrence of the criminalization of traditional Indigenous management practices and the rise of commercial fisheries. | Sharing/trading fisheries resources among community members as a mechanism for achieving equitable access to marine food. | (Whitney and Ban, 2019) |
| Coastal food resilience | The provision of food relief reduces the need for storage and preservation while causing food security deterioration, increasing dependency on disaster relief and reducing overall food security. | Reintroducing food resilience (i.e., changing the ratio of subsistence food production and tree crop commodities, revitalizing the use of famine foods, rekindling old ways and adopting new ways of preserving food crops and building on transnational kinship networks to strengthen inter-dependency food development). | (Campbell, 2015) |
| Infrastructure facilities | Increasing difficulty traveling along rivers and winter roads, as well as decreases in access to food and culturally significant species, are the challenges being faced. | Allocating resources for the procurement of advanced transportation equipment (e.g., snow machine, four-wheel all-terrain vehicle, flat-bottom or larger boat) to enhance the efficiency of coastal food distribution and increase accessibility to food. | (Lemelin et al., 2010) |
| Poverty | Indigenous inhabitants (NiVanuatu) face persistent poverty (in terms of income and risk indices), which increases resource pressure. | Demonstrating garden plots and establishing community-based reservation areas. | (Buckwell et al., 2020) |
| Communication | Weak communication networks lead to improper distribution of food. | Effectively communicating about and understanding the issue of declining fishing stock. | (Hanich et al., 2018) |
| Limited awareness | Limited awareness of climate change impacts among fishers and fishing industries will create more immediate pressures (i.e., overfishing, economic and financial limitations). | Increasing awareness by boosting the capacity to adapt and reduce risk for each fisher or fishing community. | (Lindegren and Brander, 2018) |
| Collective action | The capacity of collective action (i.e., to regulate exploitation and halt overfishing) has been diminished. | Building capacity by exploring the causal rationale among social capital and other concomitant factors affecting the enhanced/reduced adaptive capacity of small-scale fishing communities. | (Marín, 2019) |
| System changes | Coastal fisheries of most countries and territories will not meet their food security needs by 2030 because of population growth, overfishing, reduced productivity stemming from climate change and inadequate national distribution networks. | Introducing hybrid systems by incorporating elements of customary and contemporary management. | (Friedlander, 2018) |

(Buckwell et al., 2020). Constraints, contexts and possible solutions documented for constraints are given in Table 4.

## Discussion

The overarching aim of the study is to examine the constraints and limits to climate change adaptation in the context of food security among coastal Indigenous peoples. Despite the adaptation to some climate change impacts, soft and hard adaptation limits have already been seen in certain regions. For example, due to financial, governance, institutional and policy constraints, people in coastal areas of Australasia and islands, as well as small farmers from Central America, Africa, Europe and Asia, have reached soft limits leading to adverse effects on food security (IPCC, 2023, 61). Our study underlines the importance of the investigation in the context of coastal Indigenous peoples. We performed a systematic literature review with a global-level focus.

Globally, the main constraint to coastal climate change adaptation in food security settings is related to governance structures, institutional frameworks and policy limitations. Among the eight types of constraints, North America and Oceania represent a greater percentage of governance/institutions and policy constraints when analyzed by continent. Given Gibbs' (2016) observations, our findings are consistent with their conclusion that the political constraint is one of the major barriers to adaptation to climate change globally. This argument can be supplemented by the findings of Thomas et al. (2021) that, globally, the most prevalent constraints are finance, governance, institutional and policy. Our study found that climate change adaptation strategies are, in fact, influenced by a significantly larger proportion of social/cultural and economic constraints followed by financial constraints. The findings indicate that adaptation to climate change is least influenced by biological factors (such as the emergence of HABs) in coastal communities of Indigenous peoples. Since the 1980s in coastal regions, HABs have shown range expansion and increased

frequency and, thus, have negatively affected food security (Garcés and Camp, 2012). These risks are expected to become especially significant for communities with high fish consumption, that is, coastal Indigenous communities, and industry sectors such as fisheries and coastal aquaculture (Cisneros-Montemayor et al., 2016; IPCC, 2019; Galappaththi and Schlingmann, 2023). On a regional and global scale, West et al. (2021) stressed the importance of robust and more efficient HAB risk mitigation and adaptation strategies. One of our study's major findings was the identification of novel categories of constraints to climate change adaptation, such as education, communication and health.

As global warming intensifies, limits in climate change adaptation will escalate in the most vulnerable communities (Reyes-García et al., 2024a,b). This will create difficulties in avoiding these adaptation limits and signify the emergence of hard limits over soft limits. Global warming above 1.5°C could cause hard limits, indicating that ecosystems, such as warm-water coral reefs, coastal wetlands, rainforests and polar and mountain systems, will have reached or surpassed hard adaptation limits (IPCC, 2023, 61). However, our study indicates that most of the documented limits are solvable soft limits as opposed to hard limits. Coastal communities are very susceptible to climate change, and hard limits should be in place. We suspect that this discrepancy could be due to the limited documentation of hard limits in peer-reviewed articles.

The study identified ways to overcome various constraints. Such methods include improving infrastructure facilities, improving communication and awareness, building capacity and focusing on crop management strategies for coastal Indigenous communities. However, our study found very little evidence of policies addressing these constraints in coastal Indigenous communities and food security settings. For instance, Marín (2019) documented enhancing and advancing knowledge of small-scale fisheries through capacity building as a policy mechanism to regulate overfishing. As projected population growth and climate change scenarios suggest, unless measures are implemented to resolve the existing challenges, food stress might increase at a greater level than it would decrease. Thus, Campbell (2015) suggested strategies to strengthen interdependency food development (i.e., reintroducing food resilience, partly by changing the ratio of subsistence food production and tree crop commodities, revitalizing the use of famine foods, rekindling old ways of preserving food crops and adopting new ways of preserving food crops, and building transnational kinship networks). In contrast to our findings, Ford et al. (2010) revealed the positive outcomes of incorporating policy interventions in climate change adaptation constraints in Canadian Inuit populations: (i) facilitating teaching and transmission of knowledge and skills related to the environment, (ii) providing financial support for people with limited household income and (iii) increasing research efforts to identify short- and long-term risk factors and adaptive response options. IPCC (IPCC, 2023, 52) suggested that efforts to address climate change at a range of levels of governance are being accelerated by international agreements on climate change, together with increasing public awareness. Coastal adaptation planning and implementation have produced several benefits, including the potential to reduce climate risks and contribute to sustainable development through efficient adaptation options.

From a global perspective, our study results emphasize that solvable soft limits outweigh unsolvable hard limits. Among the soft limits, governance/institutions and policies stand out as the most prevalent constraints to climate change adaptation. Food security in coastal communities can be influenced by several factors (e.g., restrictions such as the absence of government support or a lack of policies to adapt to climate change) (Oulahen et al., 2018; Cabana et al., 2023; Galappaththi et al., 2024). People can be abandoned without support as a result of the absence of government programs and policies, resulting in drinking water issues, chronic food insecurity, malnutrition and hunger among low-income and marginalized communities (Chakraborty et al., 2019; Guggisberg, 2019). There is a limit to climate change adaptations resulting in food insecurity in coastal communities. As a recommendation, Whitney and Ban (2019) suggested the transformation of the existing governance model to one that recognizes Indigenous needs for social, cultural and food resources, as well as how these relate to marine resources, which will be necessary to support Indigenous peoples' ability to adapt to climate change. However, obtaining a holistic picture of the content is challenging for two reasons. One is that while we have evidence on soft limits, we lack evidence on hard limits. Thus, recommendations based solely on soft limits are not accurate. Second is that our study focused exclusively on coastal communities, which limits its ability to fully grasp the context-specific understanding.

Addressing overfishing in coastal communities demands context-specific solutions. For example, policies promoting capacity building in small-scale fisheries, as highlighted by Marín (2019), might be effective in regulating overfishing, but their implementation must align with the traditional knowledge and practices of Indigenous communities to ensure sustainability. Marín (2019) also noted that capacity building could effectively regulate overfishing in Central Southern Chile. However, different regions might require alternative approaches. For instance, governing small-scale Māori fisheries through quotas has been identified as an effective strategy for regulating overfishing (Bodwitch et al., 2024). The methods of overcoming constraints differ between Indigenous and non-Indigenous contexts, as well as between coastal and non-coastal settings. Future studies could focus on solutions discussed in the previous studies, co-designed with communities, and check whether these solutions conflict with cultural and traditional norms and values.

Climate change has become a global concern. It exerts a more significant influence on Indigenous peoples because of their strong reliance on coastal food systems, which play a crucial role in these communities (Cochrane et al., 2009; Cisneros-Montemayor et al., 2016). Successful adaptation to climate change will facilitate coastal Indigenous peoples' food security. However, emerging constraints and limits will result in maladaptations or unsuccessful adaptations, which, in turn, will influence food systems in several ways (Macintosh, 2013). Effective climate change adaptation responses positively contribute to the sustainable development of these regions (IPCC, 2023, 52). Thus, understanding the limits and constraints of climate change adaptation is essential to ensure coastal communities' food security. In adopting climate change adaptation decisions, the study can serve as a reference document to policymakers, researchers, Indigenous peoples and other relevant authorities. However, in contrast to researchers' focus on constraints linked to climate change adaptation, relatively less attention has been paid to adaptation limits, indicating similarities with the findings of Thomas et al. (2021). This creates potential avenues for future research, as we identified a gap in understanding policies aimed at addressing climate change adaptation constraints. Additionally, our study focused on the limits by dividing them into soft and hard categories and further subdividing soft limits into subcategories. Future studies can explore the different categories of soft and hard limits and examine how these terms are applied in policies to better reflect real-world scenarios.

## Conclusion

The overall aim of this study is to assess the constraints and limits associated with adaptation in terms of food security for coastal Indigenous peoples. Based on the systematic review, governance, institutions and policies are the main constraints to adaptation of climate change in coastal food security settings globally. Our study found that solvable soft limits outweigh unsolvable hard limits on a global scale. In addition, the study has identified ways of overcoming various constraints related to different contexts (i.e., improving infrastructure facilities, improving communication and awareness, building capacity and focusing on crop management strategies). We found very limited documented evidence on policies to address these constraints and limits among Indigenous peoples.

**Open peer review.** To view the open peer review materials for this article, please visit http://doi.org/10.1017/cft.2025.3.

**Supplementary material.** The supplementary material for this article can be found at http://doi.org/10.1017/cft.2025.3.

**Data availability statement.** The data will be made available upon request at any time.

**Acknowledgements.** The authors sincerely acknowledge the funding support received by the ISCE Scholars program at Virginia Tech in conducting this study.

**Author contribution.** Conceptualization: E.K.G., S.M.J.; Funding acquisition: E.K.G.; Investigation: S.M.J.; Methodology: S.M.J.; Supervision: E.K.G.; Writing – original draft: S.M.J.; Writing – review and editing: E.K.G. C.D.P., G.A.I., H.G.

**Financial support.** S.M.J., C.D.P., and G.A.I. received funds from the ISCE Scholars program hosted by the Institute for Society, Culture, and Environment at Virginia Tech.

**Competing interest.** The authors declare that they have no known competing financial interests or personal relationships that could have appeared to influence the work reported in this paper.

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
