## [Reviewer Report]

The paper is well written and well-executed. There are some syntax errors that are needed to be corrected.

E.g., Batal et al., (2021) should be corrected as Batal et al. (2021) [Remove comma].

---

## [Reviewer Report]

This is an interesting paper and embedded within it are some useful observations and findings. However, it does require quite a bit of work to make it publishable in ‘Coastal Futures’. There is repetition of material and at times a lack of focus. Structurally it could be tighter. There is a general tendency to make unsupported statements which the reader is asked to take on trust. These raise a lot of ‘why?’ and ‘how’? questions which could be solved by explaining in more detail as to how these statements were arrived at. Right at the start there is a need to define food security / insecurity so that the reader is clear on what questions are being interrogated. Fundamentally, it is not clear to me how the analysis of the 155 research articles informed the more general debate within the paper. There are some points in the Discussion where the analysis is said to challenge published views but these arguments lack clarity. It would be better to report clear findings from the study in a results section and then compare those results against the published literature. As things stand, the paper weaves throughout between the review of published papers and comments from the analysis. There are points where the paper does look at the linkages between adaptation strategies and food security but elsewhere the food security aspect gets left behind in more general text (much of it well known) on the relations between climate change and adaptation. What exactly does the analysis add to the debate? It is very important in this kind of literature scanning to give as much detail as possible on the research methods used and the decisions that were made along the way. Why were the 4 databases chosen ‘relevant’? What criteria were used to define relevance? Were other databases considered and rejected and if so why? The ‘quality’ criteria, for both screening and coding, are not adequately defined. The coding of 155 articles seems quite a low number for this kind of analysis. This leads to problems of disaggregation as then the sub-classes contain low numbers (often n = < 5) for subsequent analysis. The Discussion is over-long (by a third). There are some good critical points here but too much of the Discussion is a repetitive summary of what has gone before rather than a true discussion. The authors deserve the opportunity to revise their paper. There is good material here but major editing is needed to get their distinctive message across in what is an increasingly crowded field of literature.

---

## [Editor Report]

The authors are asked to consider revising and resubmitting an improved version of the manuscript that clarifies the contribution to the scholarship around food security and adaption. Please pay close attention to the suggestions in preparing both a revised version and a response to the reviewers document. The authors are advised to take note of Reviewer A’s suggestions to tighten the manuscript in terms of scope and purpose, especially around the definition of food security and how this paper aims to inform the knowledge landscape. In addition, a more comprehensive discussion on how decisions where made in terms of the methods is needed. More clarity on the purpose would provide a greater sense of direction for the discussion. 

Additional questions to consider for the discussion: The manuscript points to a limitation in the idea of soft and hard limits to adaptation within different local-to-global contexts but does not discuss possible nuances to this binary that may provide better insight. Would the authors recommend changes to these terms or how these limits are defined/used in a policy context? A discussion of whether or not the solutions to overcoming constraints towards food security that are presented as part of this review are feasible in different contexts - exploring the viability of solutions - would also be useful especially when carefully considering the right to self-determination for Indigenous communities. Were the solutions discussed in the reviewed papers co-designed with communities? Do some of the solutions conflict with cultural and traditional norms and values? Sense of place and attachment to place is brought up in the discussion but without context. Beyond the acknowledgement that climate change adaptation limits and constraints are significantly influenced by place attachment, there is little attempt to link this to food security. 

I look forward to seeing a revised version of the manuscript in due course.

---

## [Reviewer Report]

The revision is improved, particularly in the Methods section. The Discussion is also tighter than the original submission. I am not entirely convinced that some of the large issues have been addressed but at least they are flagged up for subsequent papers.

---

## [Editor Report]

The authors have carefully considered the reviewer comments and made worthwhile changes to the manuscript that reflect thoughtful integration of additional insights. The inclusion of a strong impact statement along with a definition for food security in the context of this manuscript is very useful. The methods section is much clearly and provides greater detail which is very welcome. The authors also raise additional areas of potential research which is useful. Understanding if the solutions raised through this global review resonate with Indigenous coastal communities, and if adoption is feasible within a range of coastal contexts, remains a key need for on-going research.